# TRABID overexpression enables synthetic lethality to PARP inhibitor via prolonging 53BP1 retention at double-strand breaks

Jian Ma [1,2,7], Yingke Zhou[2,3,7], Penglin Pan[3], Haixin Yu [4], Zixi Wang[1], Lei Lily Li[1], Bing Wang[1], Yuqian Yan[2], Yunqian Pan[2], Qi Ye[1], Tianjie Liu[1], Xiaoyu Feng[1], Shan Xu[1], Ke Wang[1], Xinyang Wang[1], Yanlin Jian[1], Bohan Ma[1], Yizeng Fan[1], Yang Gao[1], Haojie Huang [2,5,6] ✉ & Lei Li [1] ✉

53BP1 promotes nonhomologous end joining (NHEJ) over homologous recombination (HR) repair by mediating inactivation of DNA end resection. Ubiquitination plays an important role in regulating dissociation of 53BP1 from DNA double-strand breaks (DSBs). However, how this process is regulated remains poorly understood. Here, we demonstrate that TRABID deubiquitinase binds to 53BP1 at endogenous level and regulates 53BP1 retention at DSB sites. TRABID deubiquitinates K29-linked polyubiquitination of 53BP1 mediated by E3 ubiquitin ligase SPOP and prevents 53BP1 dissociation from DSBs, consequently inducing HR defects and chromosomal instability. Prostate cancer cells with TRABID overexpression exhibit a high sensitivity to poly (ADP-ribose) polymerase (PARP) inhibitors. Our work shows that TRABID facilitates NHEJ repair over HR during DNA repair by inducing prolonged 53BP1 retention at DSB sites, suggesting that TRABID overexpression may predict HR deficiency and the potential therapeutic use of PARP inhibitors in prostate cancer.

Double-strand breaks (DSBs) present a major threat to genome stability and have been linked to chromosomal translocations and cancer. DSBs are primarily repaired through homologous recombination (HR) and nonhomologous end joining (NHEJ)[1,2]. The DNA damage response (DDR) protein p53-binding protein 1 (53BP1) plays an important role in maintaining the balance between HR and NHEJ. 53BP1 promotes NHEJ by facilitating the long-range joining of broken DNA ends[3] and antagonizes HR by inhibiting DNA end resection that generates 3′ single-stranded DNA (ssDNA) overhangs necessary for the search of homologous templates[4,5]. In response to DNA damage, 53BP1 is recruited to damaged chromatin by recognizing histone H2A ubiquitylated at Lys15

(H2AK15ub) and histone H4 dimethylated at Lys20 (H4K20me2) in the nucleosome core particle[6–9], which consequently suppresses DNA end resection through interaction with PTIP and RIF1, with RIF1 recruiting to break sites the ssDNA-binding shieldin complex (REV7, SHLD1, SHLD2, and SHLD3) that protects ssDNA from degradation by end-resection nucleases[10–14].

Previous studies have implicated numerous mechanisms involved in regulating 53BP1 recruitment to the DSB sites, including proteins that compete with 53BP1 binding of H4K20me2 and H2AK15ub;[15–17] posttranslational modifications of 53BP1 that inhibit its association with chromatin;[5] direct interaction of 53BP1 tandem Tudor domain

[1]Department of Urology, The First Affiliated Hospital of Xi'an Jiaotong University, 710061 Xi'an, P. R. China. [2]Department of Biochemistry and Molecular Biology, Mayo Clinic College of Medicine and Science, Rochester, MN 55905, USA. [3]Department of Pancreatic Surgery, Union Hospital, Tongji Medical College, Huazhong University of Science and Technology, Wuhan 430022, China. [4]Cancer Center, Union Hospital, Tongji Medical College, Huazhong University of Science and Technology, Wuhan 430022, China. [5]Mayo Clinic Cancer Center, Mayo Clinic College of Medicine and Science, Rochester, MN 55905, USA. [6]Department of Urology, Mayo Clinic College of Medicine and Science, Rochester, MN 55905, USA. [7]These authors contributed equally: Jian Ma, Yingke Zhou. ✉e-mail: huang.haojie@mayo.edu; lilydr@163.com

with TIRR, which blocks the H4K20me2 binding surface of 53BP1[18,19], and so on. However, the mechanisms controlling the removal of 53BP1 from DSB sites begin to unfold. In a previous study[20], we reported that E3 ubiquitin ligase SPOP-dependent 53BP1 polyubiquitination triggers the eviction of 53BP1 from DSBs, which promotes DNA repair by HR over NHEJ. These findings imply that a deubiquitinase may enable to antagonize ubiquitination-mediated removal of 53BP1 from DSBs.

TRABID, encoded by zinc finger RANBP2-type containing 1 (*ZRANB1*) gene, belongs to the ovarian tumor (OTU) deubiquitinase (DUB) family[21]. It contains three highly conserved Npl14 zinc finger domains (3xNZF), which function as ubiquitin-binding domains (UBD), and the OTU catalytic domain responsible for the hydrolysis of ubiquitin polymers[21–23]. The crystal structure of the NZF1 domain of TRABID in complex with K29/K33 ubiquitin chains reveals a binding mode that exploits the flexibility of K29/K33 chains to achieve linkage-selective binding[24–26]. Further studies confirm that TRABID is highly specific in recognizing and processing K29 and K33 ubiquitin linkage types on several substrates, including UVRAG, JMJD2D, and HECTD1[27–29]. Besides the roles of TRABID in autophagy and transcriptional regulation, little is known about its function in DNA damage.

In the present study, we show that TRABID binds to 53BP1 at an endogenous level and deubiquitinates K29-linked polyubiquitination of 53BP1 mediated by E3 ubiquitin ligase SPOP, and prevents the dissociation of 53BP1 from DSBs, thereby promoting DNA repair by NHEJ over HR. We demonstrate that TRABID overexpression predicts HR deficiency and the potential therapeutic use of PARP inhibitors in prostate cancer.

## Results

### TRABID regulates 53BP1 foci formation at DSB sites

Posttranslational modifications (PTMs) regulate the functional assembly of 53BP1 at the DSB sites[30]. In our previous study, we showed that E3 ubiquitin ligase SPOP-mediated K29-lied polyubiquitination of 53BP1 drives the eviction of 53BP1 from DSB sites[20]. DUBs generally recognize particular ubiquitin chains[31]. To further investigate which DUB antagonizes SPOP-mediated polyubiquitination of 53BP1, we screened the DUBs that are known to cleave K29-linked ubiquitin chains, including USP16, USP19, USP32, and TRABID[26,32,33]. We knocked down each of these DUBs individually in the U2OS cell line, a cell model commonly used for the analysis of irradiation-induced foci (IRIF) formation. We found that only TRABID knockdown significantly reduced 53BP1 IRIF (Fig. 1a–c and Supplementary Fig. 1a). We further confirmed these results in prostate cancer cell line PC-3 cells (Fig. 1d–f and Supplementary Fig. 1b). These data indicate that TRABID may promote 53BP1 foci formation at DSB sites.

### TRABID binds the focus-forming region of 53BP1

To determine how TRABID regulates 53BP1 foci formation at DSB sites, we examined whether TRABID interacts with 53BP1. Co-immunoprecipitation (co-IP) assays showed that both ectopically expressed and endogenous TRABID interacted with 53BP1 in 293 T cells and PC-3 prostate cancer cells (Fig. 2a, b). To define which region in 53BP1 mediates its interaction with TRABID, we transfected 293T cells with a series of plasmids expressing hemagglutinin (HA)−tagged C-terminally truncated mutants of 53BP1 and Myc-tagged

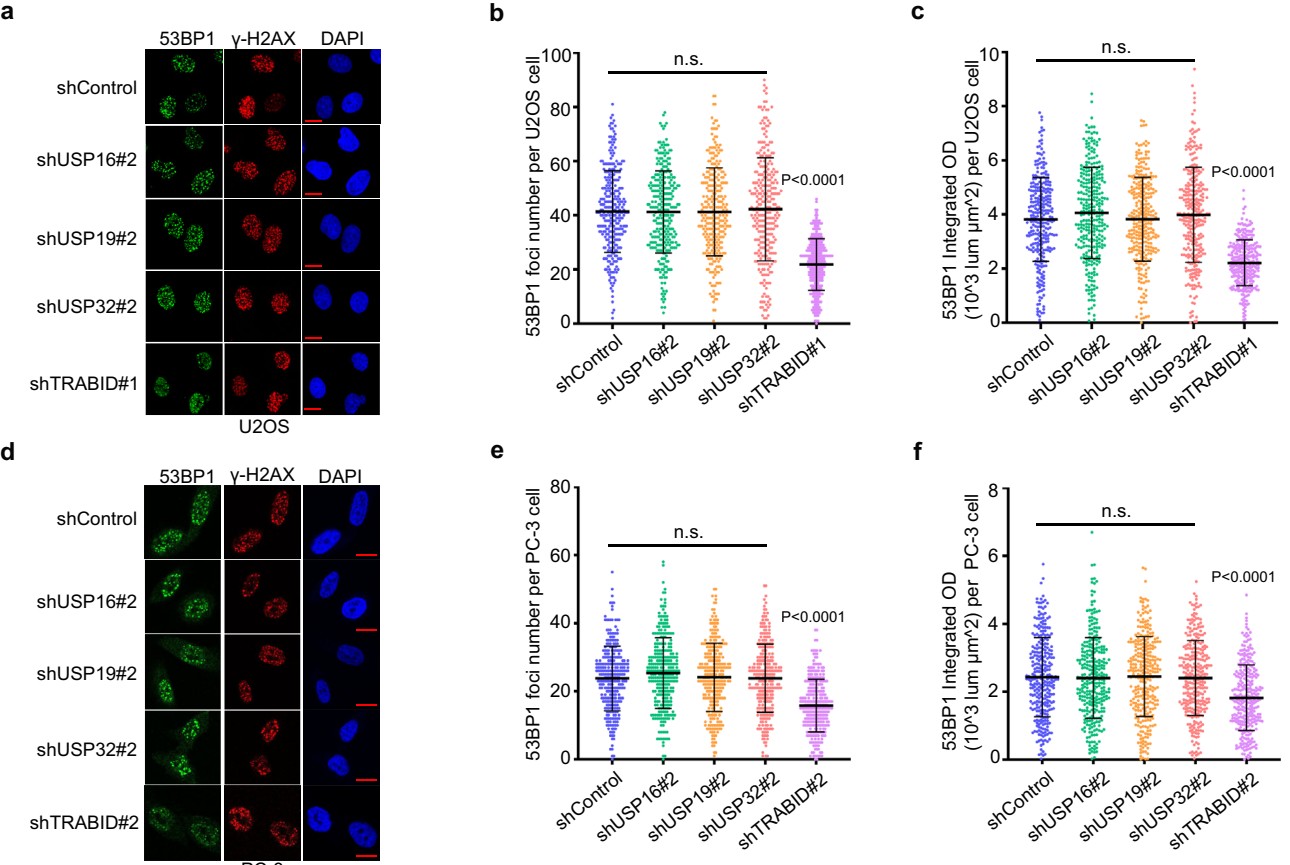

**Fig. 1 | TRABID regulates 53BP1 foci formation at DSB sites.** U2OS (**a**–**c**) and PC-3 (**d**–**f**) cells infected with lentivirus expressing control shRNAs (shControl) or shRNAs specific for indicated genes were treated with IR and subjected to 53BP1 IFC 1 h post-treatment (**a**, **d**). Scale bar, 10 μm. The average 53BP1 foci number (**b**, **e**) and foci density (**c**, **f**) in each cell were quantified. Data were presented as means ± SD of more than 300 cells from three biological replicates. n.s., not significant. Two-tailed unpaired Student's *t*-test. *P* values based on the order of appearance: **b** (0.882, 0.8655, and 0.583,1E-63); **c** (0.066, 0.9537, 0.2032, and 5.6E-47); **e** (0.0558, 0.6792, 0.9667, and 5E-27); **f** (0.8091, 0.8564, 0.7913, and 9.9E-12).

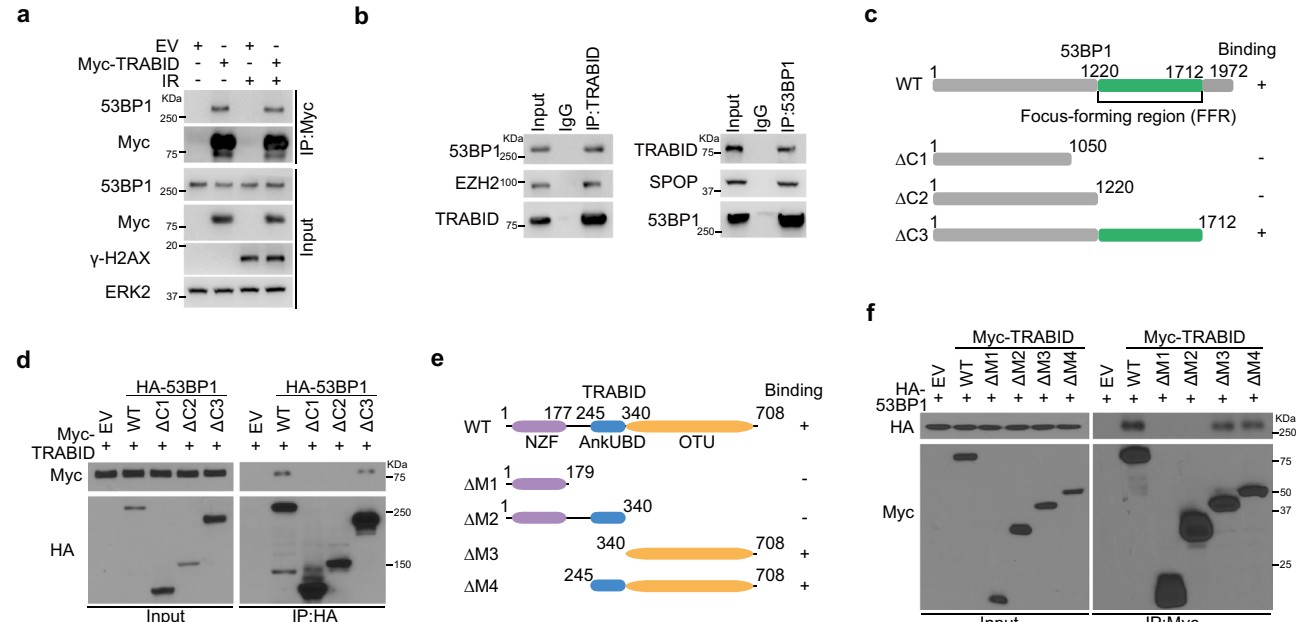

**Fig. 2 | TRABID interacts with the focus-forming region of 53BP1. a** 293T cells transfected with indicated plasmids were treated with IR for 1 h, followed by co-IP and WB. ERK2 was used as a loading control. **b** Co-IP analysis of endogenous proteins in PC-3 cells using the indicated antibodies, EZH2 and SPOP were used as positive control in each co-IP. IgG immunoglobulin G. **c** Schematic of 53BP1 domain structure and expression constructs. **d** 293 T cells transfected with indicated truncated mutants of TRABID. Co-IP assays demonstrated that the focus-forming region (FFR; residues 1220 to 1712) was required for 53BP1 binding to TRABID (Fig. 2c, d) and the OTU domain (residues 340 to 708) was necessary and sufficient for TRABID binding to 53BP1 (Fig. 2e, f). These data indicate that TRABID and 53BP1 can form a complex in cells. It is worth noting that their interaction was independent of DNA damage (Fig. 2a), suggesting that their interaction could be regulated by other factors.

constructs were harvested for co-IP and WB. **e** Schematic of TRABID domain structure and expression constructs. **f** 293T cells transfected with indicated constructs were harvested for co-IP and WB. Source data are provided in this paper or the Mendeley database (https://data.mendeley.com/datasets/n9txt6y5cj/1). Similar results for (**a, b, d, f**) panels were obtained in three independent experiments.

## TRABID antagonizes SPOP-mediated 53BP1 removal from DSB via deubiquitination of 53BP1

Since TRABID is a deubiquitinase, we examined whether TRABID regulates the protein level of 53BP1. We found that TRABID knockdown did not affect the steady-state level of 53BP1 in both PC-3 and U2OS cells regardless of IR treatment (Fig. 3a), indicating that TRABID does not regulate ubiquitin-dependent degradation of 53BP1. Notably, expression of the wild type (WT) but not catalytically inactive mutant (C443S)[24] of TRABID induced a marked decrease in 53BP1 polyubiquitination upon IR-induced DNA damage (Supplementary Fig. 2a). In contrast, expression of TRABID had little or none effect on 53BP1 deubiquitination in cells expressing SPOP F133V (Supplementary Fig. 2a). Given that SPOP specifically augments K29-linkaged polyubiquitination of 53BP1 in cells under a DNA-damaging condition and that TRABID highly recognizes and processes K29-linked ubiquitin chains[24,26], we expressed K29 residue-only ubiquitin mutant in 293T cells. Similar to the results with the WT ubiquitin, TRABID also specifically deubiquitinated K29-linked polyubiquitination of 53BP1 in SPOP WT but not F133V mutant cells (Fig. 3b and Supplementary Fig. 2b). Moreover, instead of slowly increased K29-linked polyubiquitination of 53BP1 after IR treatment (peaks at 4 h), TRABID knockdown lead to a rapid K29-linked polyubiquitination of 53BP1 shortly after IR treatment (peaks at 1 h) (Fig. 3c and Supplementary Fig. 2c). These data indicate TRABID antagonizes SPOP-mediated 53BP1 polyubiquitination.

To understand how TRABID-mediated 53BP1 deubiquitination influences 53BP1 foci at DSB sites, we infected PC-3 and U2OS cells with lentivirus expressing control shRNA or TRABID-specific shRNA in combination with empty vector (EV) or SPOP F133V mutant (Supplementary Fig. 2d). While TRABID knockdown decreased 53BP1 IRIF at 1, 4, and 8 h after IR, SPOP F133V mutation totally reversed 53BP1 retention at DSB sites enhanced by TRABID depletion (Fig. 3d–f and Supplementary Fig. 2e–g). Notably, the effect of TRABID on 53BP1 deubiquitination and retention at DSB sites are unlikely caused by the impact of TRABID on SPOP binding to 53BP1 since neither expression of TRABID WT, nor the enzymatic dead mutant C433S had any obvious effect on SPOP-53BP1 interaction (Supplementary Fig. 2h). These data indicate that TRABID antagonizes SPOP-mediated 53BP1 removal from DSBs via deubiquitinating 53BP1.

## TRABID is transcriptionally repressed by the NuRD complex after IR treatment

In different in vivo ubiquitination assays performed in 293T cells, we invariably noticed that expression of endogenous TRABID protein was downregulated following IR treatment (Fig. 3c and Supplementary Fig. 2c). We further confirmed that this was the case in both PC-3 and U2OS cells without ubiquitin transfection (Fig. 4a). Notably, DNA damage-induced downregulation of TRABID expression also occurred at the mRNA level in both PC-3 and U2OS cell lines (Fig. 4b), but IR treatment had little or no effect on the level of the ectopically expressed Myc-TRABID (Fig. 2a). Together, these data suggest that TRABID is likely transcriptionally repressed after DNA damage.

Previous studies show that the nucleosome remodeling and deacetylation (NuRD) complex is recruited to the DSB sites and mediates transcriptional repression after DNA damage[34–37]. We, therefore, performed a meta-analysis of published ChIP-seq datasets and found that several NuRD complex components, including LSD1, MBD2, MBD3, GATAD2B, and HDAC1, bind to the same region proximal to the promoter of the *ZRANB1* gene (Supplementary Fig. 3). To experimentally

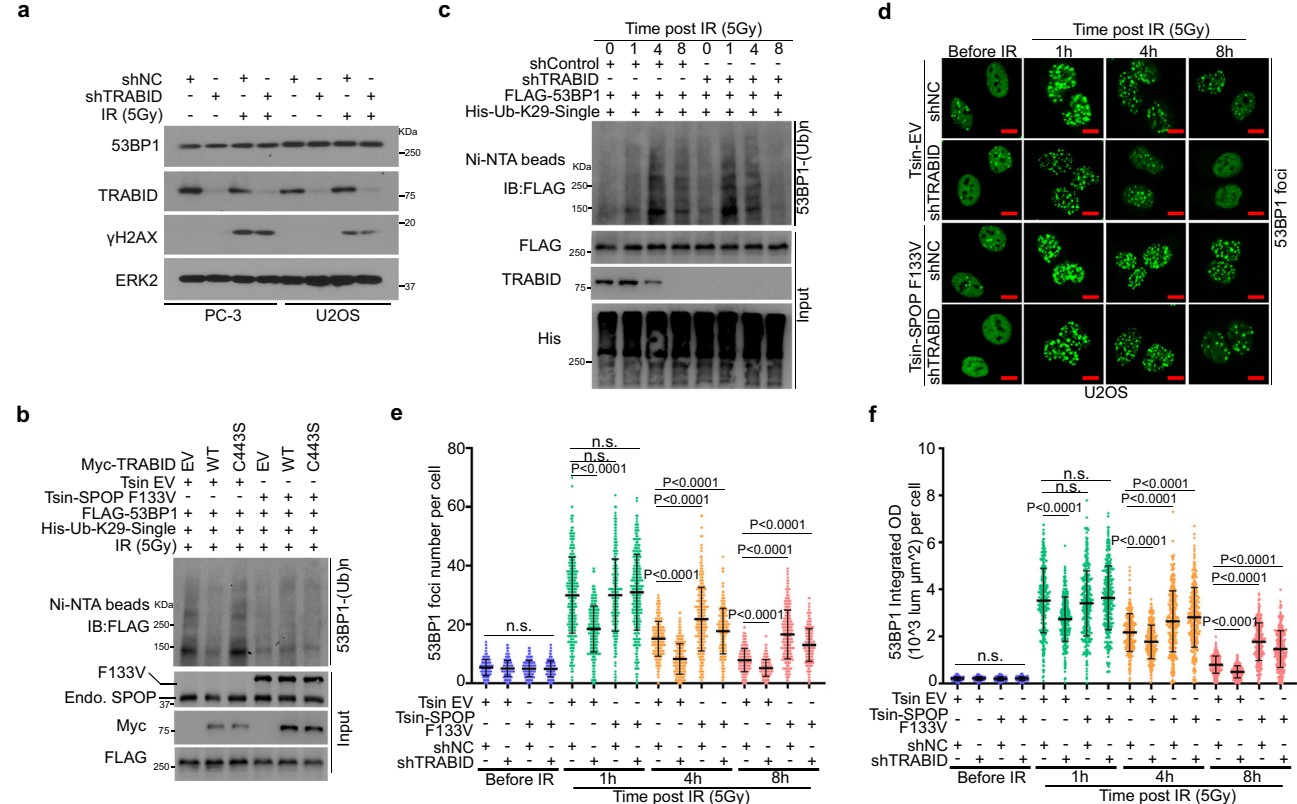

**Fig. 3 | TRABID antagonizes SPOP-mediated removal of 53BP1 from DSB via deubiquitination of 53BP1. a** PC-3 and U2OS cells infected with lentivirus expressing indicated shRNAs were treated with IR for 1 h and harvested for WB analysis. **b** Ubiquitination assay using Ni-NTA pull-down in 293T cells transfected with His-K29-single ubiquitin together with other indicated plasmids and treated with IR (5 Gy) for 1 h. **c** Ubiquitination assay using Ni-NTA pull-down in 293T control and TRABID knockdown cells transfected with indicated plasmids. Cells were harvested at different time points after IR. **d**–**f** U2OS cells infected with lentivirus expressing indicated shRNA were treated with IR followed by IFC of 53BP1 at the

indicated time points after IR (**d**). Scale bar, 10 μm. The average 53BP1 foci number (**e**) and foci density (**f**) in each cell were quantified. Data were presented as means ± SD of more than 300 cells from three biological replicates. Two-tailed unpaired Student's *t*-test. n.s., not significant. *P* values based on the order of appearance: **e** (0.154, 0.08, 0.054, 3E-35, 0.9716, 0.3365, 1.9E-44, 1.3E-19, 5.1E-6, 5.3E-20, 1.9E-50, and 2.3E-34); **f** (0.424, 0.8355, 0.9635, 5.8E-16, 0.2608, 0.34, 1.6E-10, 4.4E-8, 2.6E-13, 7.2E-29, 1.1E-62, and 3.5E-35). Source data are provided in this paper or the Mendeley database (https://data.mendeley.com/datasets/n9txt6y5cj/1). Similar results for (**a**–**c**) panels were obtained in two independent experiments.

validate this observation, we carried out the Cleavage Under Targets and Tagmentation (CUT&Tag) assay, an enzyme-tethering strategy that provides efficient high-resolution sequencing libraries for profiling diverse chromatin components[38,39]. Both the LSD1 and MBD3 CUT&Tag profiling showed increased binding of these two proteins at the *ZRANB1* gene locus after IR treatment (Fig. 4c, d), and importantly, the binding region is the same as the site shown by the ChIP-seq data (Supplementary Fig. 3). The IR-enhanced binding of LSD1 and MBD3 proteins in this genomic locus was further confirmed by ChIP-qPCR in PC-3 and U2OS cells (Fig. 4e, f). To experimentally verify whether TRABID is a repression target of NuRD complex after DNA damage, we examined the effect of LSD1 and MBD3 on TRABID expression at both mRNA and protein levels. Knockdown of endogenous LSD1 and MBD3 abolished IR-induced TRABID downregulation in both PC-3 and U2OS cells (Fig. 4g–j). These data indicate that TRABID expression is downregulated by the NuRD complex after IR treatment.

A temporal analysis in different cell types revealed that 53BP1 IRIF were most prominent during the G1 phase of the cell cycle and progressively decreased in intensity when cells transited from early to late S phases[40]. We sought to determine whether TRABID regulates 53BP1 deubiquitination in a cell cycle-dependent manner. To this end, we synchronized PC-3 and U2OS cells by nocodazole treatment and release (Fig. 4k). We found that TRABID protein expression varied during the cell cycle with the lowest expression occurring in the S phase (Fig. 4l). Similar to protein expression, we further demonstrate that both *ZRANB1* mRNA expression also oscillated during the cell

cycle with low expression at G1 and S phases in PC-3 cells (Fig. 4m–p). The level of the NuRD complex components LSD1 and MBD3 peaked in the early S phase as previously reported[41] (Fig. 4m, o), and knockdown of LSD1 or MBD3 abolished the oscillation of TRABID expression level (Fig. 4m–p). These data imply that these components of the NuRD complex play an important role in the cell cycle-dependent regulation of TRABID expression. In agreement with the expression pattern of TRABID during the cell cycle, we found that K29-linked polyubiquitinated of 53BP1 was peaked in the S phase of PC-3 cells; however, depletion of TRABID delayed the process of deubiquitination of 53BP1 during the cell cycle (Fig. 4q). These observations indicate that TRABID is transcriptionally repressed by the NuRD complex in response to DNA damage, bestowing K29-linked polyubiquitination and removal of 53BP1 from DSB sites.

## TRABID overexpression inhibits HR activity and promotes chromosomal instability

It is reported that TRABID protein level was correlated with poor survival in breast cancer[42], but the expression status of TRABID in prostate cancer is barely known. We examined TRABID expression in a cohort of prostate cancer patient specimens (*n* = 30) using immunohistochemistry (IHC). TRABID expression was significantly higher in prostate tumor tissues compared to adjacent normal tissues (Fig. 5a–c), indicating TRABID is overexpressed in prostate cancer. To mimic the pathophysiological conditions in patients, we stably expressed TRABID WT and the catalytically inactive mutant C443S[24] (a negative

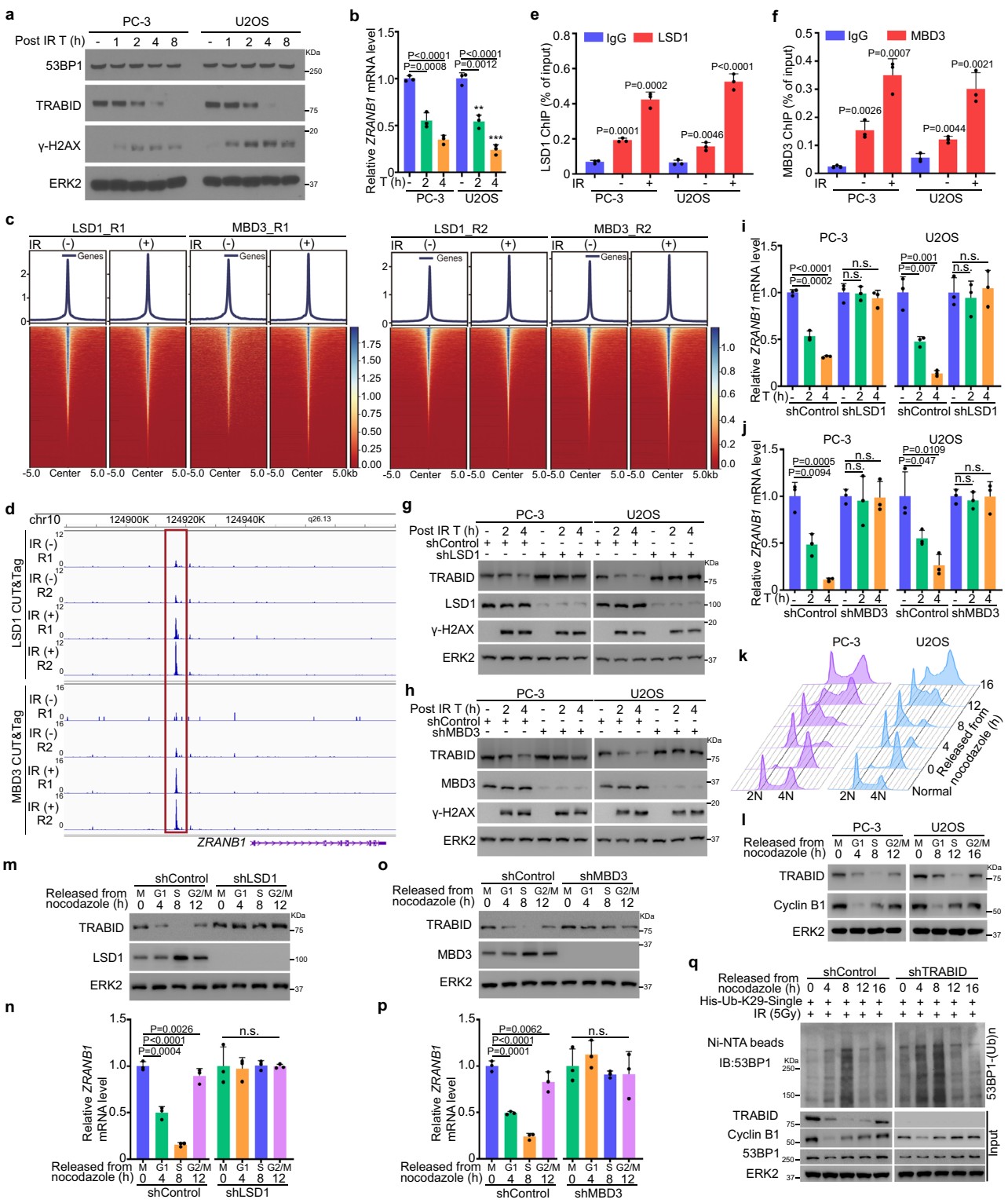

control) into PC-3 and U2OS cell lines (Supplementary Fig. 4a). As expected, TRABID expression prolonged 53BP1 retention at DSB sites (Fig. 3d–f and Supplementary Fig. 3e–g); however, expression of TRABID WT but not C443S significantly decreased the number of IRIF of HR mediators BRCA1 and RAD51 (Fig. 5d–g and Supplementary Fig. 4b–i). Of note, knockdown of 53BP1 abolished TRABID-mediated decrease in BRCA1 and RAD51 foci (Fig. 5d–g and Supplementary Fig. 4b–i), suggesting that TRABID overexpression-mediated inhibition of HR protein foci formation is 53BP1-dependent. Moreover, we

showed that the expression of TRABID WT but not catalytically inactive mutant C443S decreased the HR reporter activity, whereas it increased the NHEJ reporter activity in PC-3 cells (Fig. 5h, i). Furthermore, camptothecin (CPT) treatment resulted in a marked increase in the number of chromosomal breaks per cell in TRABID WT-expressing but not TRABID-C443S-expressing PC-3 cells (Fig. 5j, k). These data indicate that TRABID is overexpressed in prostate cancer patient specimens we examined, and TRABID overexpression inhibits HR activity and induces chromosomal instability under DNA damage conditions.

**Fig. 4 | TRABID is transcriptionally repressed by the NuRD complex after IR.**
**a** PC-3 and U2OS cells were treated with IR and harvested at the indicated time points (T) after IR for WB analysis. **b** PC-3 and U2OS cells were treated with or without IR and harvested at the indicated time points after IR for RT-qPCR analysis. Data were shown as the mean ± SD of three independent experiments (*n* = 3). Two-tailed unpaired Student's *t*-test. *P* values based on the order of appearance: 0.0008, 3.3E-5, 0.0012, and 9.5E-5. **c** Heatmaps show LSD1 and MBD3 CUT&Tag profiling sequencing read intensity throughout the genome in PC-3 cells with or without IR treatment. **d** CUT&Tag signal tracks showing LSD1 and MBD3 occupancy at *ZRANB1* gene locus in PC-3 cells with or without IR treatment. **e, f** ChIP-qPCR analysis of LSD1 (**e**) and MBD3 (**f**) occupancy at the *ZRANB1* gene locus in both PC-3 and U2OS cells with or without IR treatment. Data were shown as the mean ± SD of three independent experiments (*n* = 3). Two-tailed unpaired Student's *t*-test. *P* values based on the order of appearance: in **e** (0.0001, 0.0002, 0.0046, and 7.3E-5) and **f** (0.0026, 0.0007, 0.0044, and 0.0021). **g–j** PC-3 and U2OS cells were infected with lentivirus expressing indicated shRNAs and harvested at the indicated time points (T) after IR for WB analysis (**g, h**) and RT-qPCR analysis (**i, j**). Data were shown as the mean ± SD of three independent experiments (*n* = 3). Two-tailed unpaired Student's *t*-test. *P* values based on the order of appearance: in **i** (0.0002, 2.7E-6, 0.91, 0.4781,

0.0072, 0.001, 0.7017, and 0.7798) and in **j** (0.0094, 0.0005, 0.7659, 0.8925, 0.047, 0.0109, 0.5247, and 0.9567). **k, l** PC-3 and U2OS cells were synchronized with nocodazole (100 ng/ml) for 12 h and released into the cell cycle. At the indicated time points (T), cells were harvested for cell cycle analysis by FACS (**k**) and WB analysis (**l**). **m–p** PC-3 cells infected with lentivirus expressing indicated shRNAs were synchronized with nocodazole (100 ng/ml) for 12 h and released into the cell cycle. At the indicated time points, cells were harvested for WB analysis (**m, o**) and RT-qPCR analysis (**n, p**). Data were shown as the mean ± SD of three independent experiments (*n* = 3). n.s. not significant. *P* values based on the order of appearance: in **n** (0.0004, 1E-5, 0.0026, 0.8436, 0.9526, and 0.7412) and in **p** (0.0001, 3.6E-5, 0.0062, 0.4242, 0.4398, and 0.2789). **q** PC-3 cells were infected with lentivirus expressing indicated shRNAs transfected with His-Ub-K29-single construct and released from nocodazole treatment. Cells were harvested for ubiquitination assay using Ni-NTA pull-down at indicated time points. The sequencing data for CUT&Tag have been deposited to the GEO database with the accession code GSE222267. Source data are provided in this paper or the Mendeley database (https://data.mendeley.com/datasets/n9txt6y5cj/1). Similar results for (**a, g, h, l, m, o, q**) panels were obtained in two independent experiments.

## TRABID overexpression enables synthetic lethality to PARP inhibitors in prostate cancer cells

TRABID overexpression in prostate cancer patient samples and TRABID overexpression-mediated HR activity inhibition and chromosomal instability prompted us to determine whether overexpressed TRABID enables synthetic lethality to PARP inhibitor in prostate cancer cells. A dose course study in both PC-3 and U2Os cells revealed that the IC50 of the PARP inhibitor olaparib in TRABID WT-expressing cells was much lower compared with EV control and TRABID-C443S-expressing cells; however, this effect of TRABID WT was abolished by 53BP1 knockdown (Fig. 6a, b and Supplementary Fig. 4a). By performing colony formation assays, we confirmed that olaparib markedly inhibited the growth of TRABID WT-expressing cells with formation of fewer and smaller colonies, but only minimal effect was observed in EV control and TRABID-C443S-expressing cells (Fig. 6c–f). We further investigated this effect in vivo. PC-3 xenograft tumors were generated by subcutaneous injection of PC-3 cells into SCID mice. By treating mice with vehicle control and PARP inhibitor, we demonstrated that olaparib significantly inhibited tumor growth of TRABID WT-expressing tumors compared with EV control and TRABID-C443S-expressing tumors (Fig. 6g, h). These data indicate that TRABID overexpression sensitizes prostate cancer cells to PARP inhibitors in vitro and in vivo.

## Discussion

Posttranslational modifications (PTMs) control the functional assembly and disassembly of numerous proteins and PTMs play important roles in regulating 53BP1 recruitment to DNA damage sites. 53BP1 is rapidly recruited to damaged chromatin via multivalent interactions with histone PTMs, including H4K20me2 and H2AK15ub[6,8]. This process is further regulated by factors like JMJD2A, L3MBTL1, RAD18, RNF169, and TIRR that restrict the 53BP1 reading of these PTMs[19,43–49]. 53BP1 is excluded from DSB sites during the G1-S transition of the cell cycle while BRCA1 recruitment is elevated[40], implying a temporal NHEJ-to-HR switch after chromatin engagement of 53BP1[40]. However, the molecular mechanism underlying this transition from NHEJ to HR had not been fully elucidated. Previously, we have reported that SPOP binds to 53BP1 in the S phase of the cell cycle and promotes K29-linked polyubiquitination of 53BP1, which leads to 53BP1 extraction from DSB sites[20]. We provide evidence in the present study that the deubiquitinase TRABID interacts with 53BP1 at the endogenous level. We further show that TRABID is overexpressed in prostate cancer, and its overexpression antagonizes SPOP-mediated K29-linked polyubiquitination of 53BP1, which consequently prolongs the retention of 53BP1 at the DSB sites and results in the selection of NHEJ over HR after chromatin engagement of 53BP1 (Fig. 7).

To ensure accurate DNA repair, it is necessary to maintain the dynamic equilibrium between key factors like 53BP1 and BRCA1 at the DSB sites in normal cells, for which both ubiquitination and deubiquitination play important roles in maintaining this balance. We have shown that TRABID antagonizes SPOP-mediated 53BP1 polyubiquitination, which consequently blocks SPOP-induced 53BP1 removal from DSB sites. However, these would not happen under normal conditions because TRABID is transcriptionally repressed by the NuRD complex after DNA damage, conceding to SPOP-mediated 53BP1 removal from DSB sites. In prostate cancer cells with TRABID overexpression, however, high-level expression of TRABID overrides SPOP-mediated 53BP1 polyubiquitination, which consequently prolongs 53BP1 retention at DSB sites and inhibits HR activity.

PARP inhibitors (PARPi) have been approved for the treatment of men with metastatic castration-resistant prostate cancer (mCRPC) with selected HR deficiencies (BRCA1/2 or ATM mutations)[50,51]. Apparently, there are many other mechanisms leading to HR deficiencies. However, all the HR deficiencies targeted for cancer treatment are genetic alterations, which precludes other mechanisms driving HR deficiencies, which include protein posttranslational modifications. Our findings in the present study show that TRABID overexpression contributes to HR deficiencies and confers synthetic lethal to PARP inhibitor in prostate cancer cells (Fig. 7). Thus, our findings suggest that in addition to the genetic alterations PTMs of DNA repair modulators can play a pivotal role in HR deficiencies and PARP inhibitor sensitivity.

In summary, we demonstrate that TRABID plays an important role in regulating 53BP1 retention at DSB sites by deubiquitinating K29-linked polyubiquitination. Under normal conditions, TRABID expression is transcriptionally repressed by the NuRD complex in response to DNA damage, which facilitates SPOP-mediated 53BP1 removal from DSB sites. We further show that TRABID is overexpressed in prostate cancer and TRABID overexpression inhibits HR activity and promotes chromosomal instability. Accordingly, we demonstrate that TRABID overexpression enables synthetic lethality to PARP inhibition in prostate cancer, stressing that TRABID overexpression represents a vulnerability to PARP inhibitors in cancers such as prostate cancer.

## Methods

Patient sample analysis in this study was approved by the ethics committee of The First Affiliated Hospital of Xi'an Jiaotong University (Xi'an, China). All patients provided written informed consent to participate in the study.

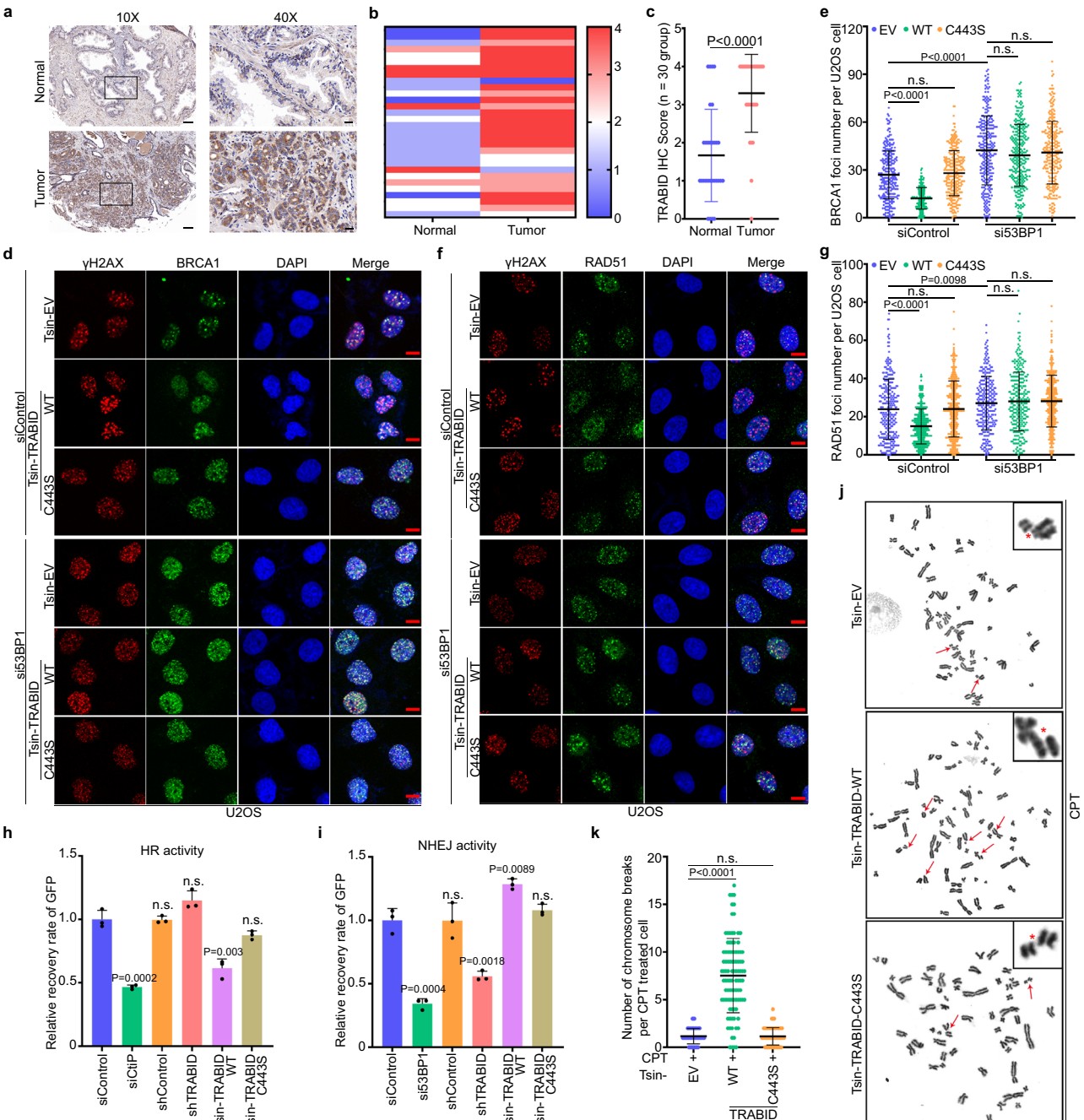

**Fig. 5 | TRABID overexpression inhibits HR activity and promotes chromosomal instability. a–c** Representative images of IHC staining (**a**) of TRABID protein on paired prostate cancer patient specimens (*n* = 30). Scale bar in 10 X fields: 100 μm; Scale bar in 40 X fields: 20 μm. Heat map showing TRABID IHC score in normal and paired tumor tissues (see calculation details in Methods) (**b**). Quantification of IHC score (**c**). Data were shown as the mean ± SD of three independent experiments (*n* = 3). Two-tailed unpaired Student's *t*-test. *P* = 5.3E-7. **d–g** U2OS cells infected with lentivirus expressing EV, TRABID-WT, or TRABID-C443S and treated with siControl or si53BP1 were exposed to IR followed by IFC of BRCA1 (**d**) and RAD51 (**f**) at 1 h after IR. Scale bar, 10 μm. The average foci number (**e, g**) in each cell were quantified. Data were presented as means ± SD of more than 300 cells from three biological replicates. Two-tailed unpaired Student's *t*-test. n.s. not significant. *P* values based on the order of appearance: **e** (2.5E-54, 0.3363, 1.9E-22, 0.0659, and 0.4015), **g** (1.9E-16, 0.9275, 0.0098, 0.4221, and 0.3285). **h, i** PC-3 cells were

transfected with HR or NHEJ reporter in combination with TRABID WT/C443S or small interfering RNA (siRNA) for CtIP or 53BP1 and HR (**h**) and NHEJ (**i**) activities were measured. Data were represented as means ± SD of three biological replicates. Two-tailed unpaired Student's *t*-test. n.s. not significant. GFP green fluorescent protein. *P* values based on the order of appearance: **h** (0.0002, 0.9571, 0.0711, 0.003, and 0.0537), **i** (0.0004, 0.9819, 0.0018, 0.0089, and 0.2552). **j, k** PC-3 cells infected with lentivirus expressing EV, TRABID-WT, or TRABID-C443S were treated with CPT (1 μM) for 24 h. Cells were harvested for karyotyping, and chromosome breaks in more than 80 cells from three biological replicates in each group were counted and quantified. Representative images were shown in (**j**) and quantitative data are shown in (**k**). Data were shown as the mean ± SD of three independent experiments (*n* = 3). Two-tailed unpaired Student's *t*-test. *P* values based on the order of appearance:1.3E-37, 0.9348. n.s. not significant. Source data are provided in this paper.

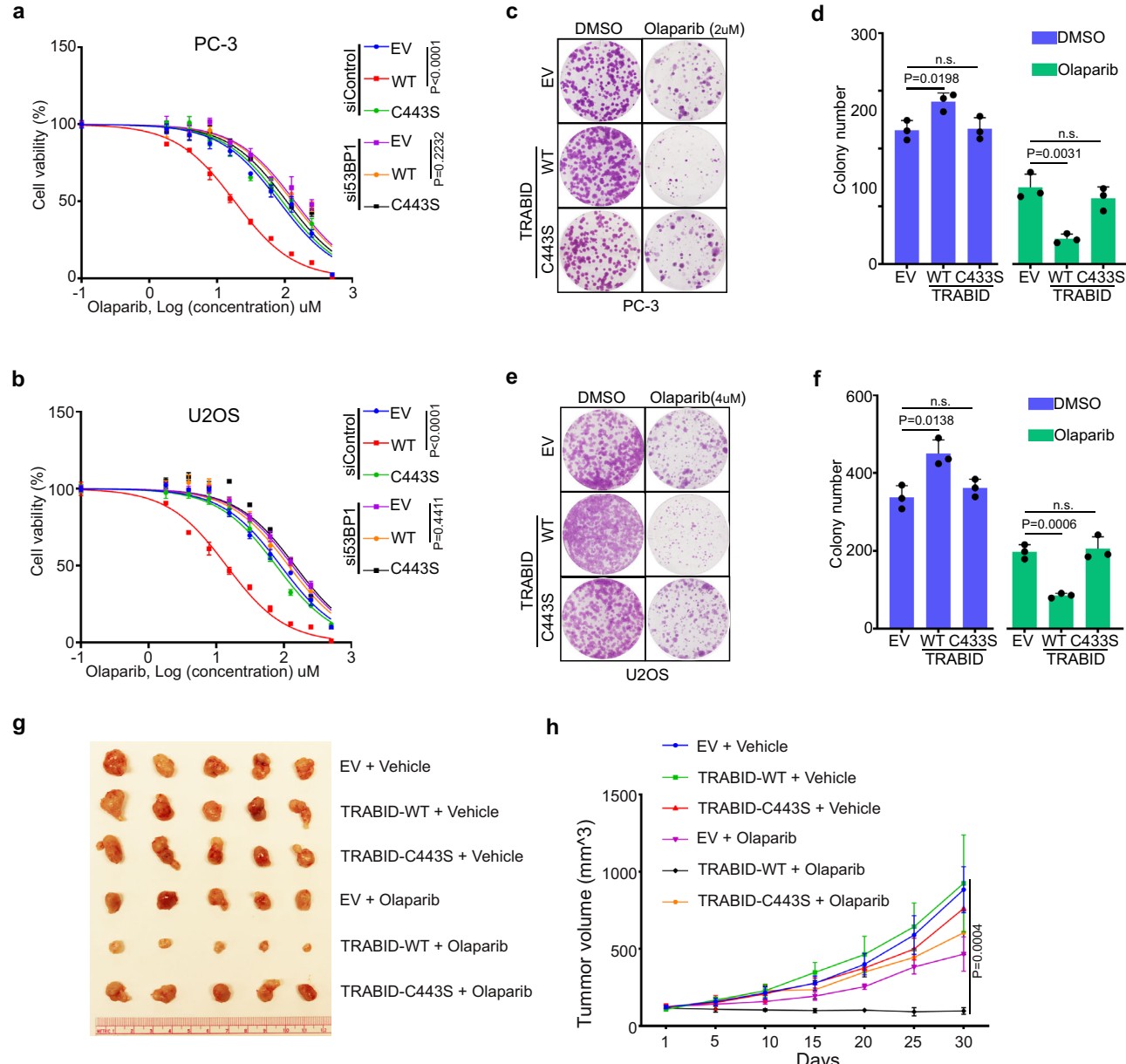

**Fig. 6 | TRABID overexpression enables synthetic lethality to PARP inhibitor in prostate cancer cells. a**, **b** Dose-response survival curves of EV, TRABID-WT, and TRABID-C443S-expressing cells combine with siControl or si53BP1 exposed to increasing concentrations of olaparib in PC-3 (**a**) and U2OS (**b**) cells. Data were shown as the mean ± SD of three independent experiments ($n = 3$). Statistical analysis was performed using two-way ANOVA. $P$ value were provided in the figure. **c**–**f** Colony formation assays were performed in PC-3 (**c**, **d**) and U2OS (**e**, **f**) cell lines infected with lentivirus expressing EV, TRABID-WT, or TRABID-C443S. The number of colonies was counted. Representative colonies are shown in (**c**, **e**), with

quantification data shown in (**d**, **f**). Data were presented as the mean ± SD of three independent experiments. Two-tailed unpaired Student's $t$-test. $P$ value were provided in the figure. **g**, **h**. PC-3 cells infected with lentivirus expressing EV, TRABID-WT, or TRABID-C443S were injected s.c. into the right flank of SCID mice and treated with vehicle or olaparib (50 mg/kg). Tumor growth was measured every five days for 30 days. Tumors in each group at day 30 were harvested, photographed and shown in (**g**). Data in (**h**) are shown as mean ± SD ($n = 5$). Two-tailed unpaired Student's $t$-test. $P$ value were provided in the figure. Comparing the size of tumors in different groups at day 30. Source data are provided in this paper.

## Cell lines, cell culture, and transfection

PC-3, U2OS, and 293T cells were obtained from the American Type Culture Collection (Manassas, VA). 293T and U2OS cells were cultured in Dulbecco's modified Eagle's medium (DMEM) supplemented with 10% of FBS (Thermo Fisher Scientific). PC-3 cells were cultured in RPMI 1640 medium supplemented with 10% FBS. The cells were maintained in a 37 °C humidified incubator supplied with 5% $CO_2$. Transient transfection was performed by Lipofectamine 2000 (Thermo Fisher Scientific). Lentiviral shRNA constructs were transfected using the Calcium Phosphate protocol. pTsin-TRABID-WT or pTsin-TRABID-

C443S mutant expression and virus packing constructs were transfected into 293 T cells. The virus supernatant was collected 48 h after transfection. Target cells were infected with viral supernatant in the presence of polybrene (8 μg/ml) and were then selected in growth media containing 1.5 μg/ml puromycin. All the cell lines used have been tested and authenticated by karyotyping, and prostate cancer cell lines have also been authenticated by examining TRABID status. Plasmocin (InvivoGen) was added to cell culture media to prevent mycoplasma contamination. Mycoplasma contamination was tested regularly using the Lookout Mycoplasma PCR Detection Kit from Sigma-Aldrich.

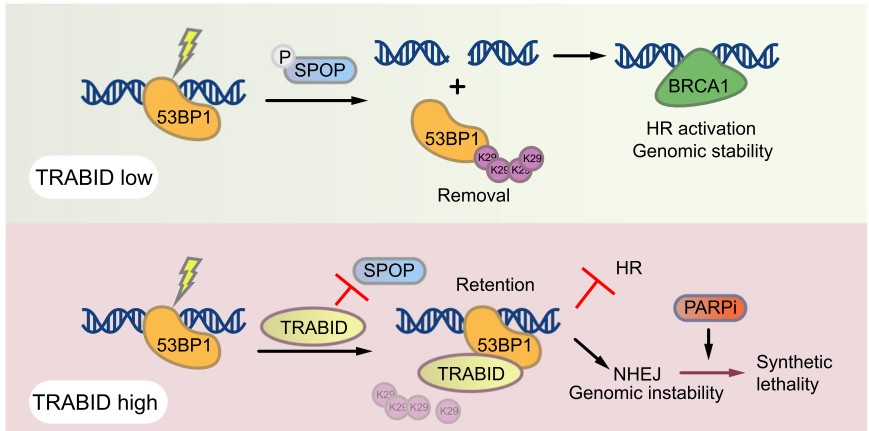

**Fig. 7 | Working model illustrating how TRABID regulates genome stability by promoting 53BP1 retention at DSB sites and selection of HR over NHEJ and enabling synthetic lethality to PARP inhibition.** Under normal conditions, TRABID expression is low and SPOP is responsible for the removal of 53BP1 from DSB sites, thereby promoting HR over NHEJ. However, prostate cancer-associated TRABID overexpression promotes the deubiquitination of SPOP-dependent K29-linked polyubiquitination of 53BP1, which consequently leads to the prolonged retention of 53BP1 at the DSB sites, resulting in the selection of NHEJ over HR after 53BP1 engagement with chromatin at DSB sites and enabling synthetic lethality to PARP inhibition. K29 with a circle, K29-linked ubiquitin.

## Antibody information
Primary antibodies used include TRABID (Abcam, # ab262879, 1:1000), 53BP1 (Abcam, # ab36823, 1:1000), HA.11 (Covance, #MMS-101R, 1:1000), SPOP (Proteintech Group, # 16750-1-AP, 1:1000), BRCA1 (Santa Cruz, # sc-642, 1: 500), RAD51 (Santa Cruz, # sc-377467, 1: 500), Myc (Santa Cruz, # sc-40, 1:1000), Flag (Sigma, # F-3165, 1:1000), ERK2 (Santa Cruz, # sc-1647, 1:1000), Phospho histone H2A.X (S139) (Cell Signaling, # 9718, 1:1000), Phospho histone H2A.X (S139) (Cell Signaling, # 80312 S, 1:1000), LSD1 (Abcam, # ab129195, 1:1000), MBD3 (Abcam, # ab157464, 1:1000), Second antibodies were Rabbit IgG (H + L) Alexa Fluor 594 (Thermo Fisher, # A11037, 1:500), Rabbit IgG (Jackson ImmunoResearch, # 211-032-171, 1:5000), Mouse IgG (Jackson ImmunoResearch, #115-035-174, 1:5000), Mouse IgG (H + L) Alexa Fluor 488 (Thermo Fisher, # A11029, 1:500), Rat IgG (H + L) Alexa Fluor 488 (Life Technologies, # A-11006, 1:500), and Mouse IgGκ BP-FITC (Santa Cruz, # sc-516140, 1:500).

## RNA interference and sgRNA-mediated gene deletion
Nonspecific control small interference RNA (siRNA) and siRNAs for CtIP and 53BP1 were purchased from GE Dharmacon. The sequences of siRNA oligos are as follows: siCtIP#1 5′-GCUAAAACAGGAACGAAUC-3′ and siCtIP#2 5′-UCCACAACAUAAUCCUAAU-3′; si53BP1#1 5′-GAGCUGGGAAGUAUAAAUU-3′ and si53BP1#2 5′-GGACUCCAGUGUUGUCAUU-3′.

siRNA transfection of cells was performed following the manufacturer's instructions. The lentivirus-expressing shRNAs were purchased from Sigma-Aldrich and performed following the manufacturer's instructions.

## Co-immunoprecipitation (co-IP)
Cells were harvested and lysed by IP buffer (50 mM Tris-HCl, pH 7.4, 150 mM NaCl, 1% Triton X-100, 1% sodium deoxycholate, and 1% protease inhibitor cocktails) on ice for more than 15 min. The cell lysate was centrifuged for 15 min at 18,000×g at 4 °C, and the supernatant was incubated with primary antibodies and protein A/G agarose beads (Thermo Fisher Scientific) with rotating at 4 °C overnight. The next day, the pellet was washed at least six times with 1×IP buffer on ice and then subjected to western blotting analysis.

## Immunofluorescent chemistry (IFC) and foci quantification
PC-3 or U2OS cells were seeded on 13-mm glass coverslips. After washing once in ice-cold phosphate-buffered saline (PBS), the coverslips were fixed with 4% formaldehyde in PBS for 15 min at room temperature. Before incubation with primary antibodies, the coverslips were permeabilized in 0.5% Triton in PBS for 15 min. The samples were incubated with primary antibodies for 2 h, washed at least three times with 0.01% Tween 20 in PBS (PBS-T), and subsequently incubated with secondary fluorescence-coupled antibodies for 1 h. After three times' washing, incubations with antibodies were performed in 3% bovine serum albumin in PBS-T. The coverslips were stained with 4′,6-diamidino-2-phenylindole at 10 mg/ml for 5 min before being mounted on glass slides and visualized using a fluorescence microscope. DNA damage foci were quantified using the Image-Pro software. For each cell line analyzed with Image-Pro, 10 randomly picked photographs that included more than 200 cells were used to standardize foci counting and integration of optical density. This approach was used to analyze approximately 300 cells for each series of experiments.

## Ubiquitination assay
For ubiquitination using Flag-conjugated agarose beads or Flag primary antibody plus protein A/G beads, 293T cells were transfected with plasmids for HA-Ub, Flag-53BP1, and other indicated constructs. Cells were harvested and lysed with lysis buffer (50 mM Tris-HCl, pH 7.5, 150 mM NaCl, 1% NP40, 0.5% sodium deoxycholate, and 1× protease inhibitor cocktail (PIC)). The lysate was subjected to co-IP using anti-Flag-conjugated agarose beads or Flag primary antibody plus protein A/G beads.

For ubiquitination using nickel-nitrilotriacetic acid (Ni-NTA) beads (QIAGEN), cells were transfected with His-ubiquitin and indicated constructs for 42 h. Subsequently, cells were lysed with buffer A (6 M guanidine-HCl, 0.1 M Na2HPO4/NaH2PO4, and 10 mM imidazole [pH 8.0]) and sonicated for 15 s. After incubating with nickel-nitrilotriacetic acid (Ni-NTA) beads (QIAGEN) for 3 h at room temperature, the proteins were washed twice with buffer A, twice with buffer A/TI (1 volume buffer A and 3 volumes buffer TI), and one time with buffer TI (25 mM Tris-HCl and 20 mM imidazole [pH 6.8]). The pull-down proteins were denatured at 95 °C for 5 min and separated by SDS-PAGE for immunoblotting.

## Tissue microarray immunohistochemistry (IHC)
The prostate cancer tissue microarray (TMA) slides were generated by a total of 30 pairs of prostate cancer and tumor-adjacent specimens, which were obtained from the First Affiliated Hospital of Xi'an Jiaotong

University (Xi'an, China) and approved by the Ethical Committee of the Hospital. Informed consent was obtained from all patients. The TMA slides were stained with TRABID antibody (1:1000 dilution) by standard immunohistochemistry procedures. Images were acquired using a Leica microscope. IHC staining intensity and staining percentage were graded as follows: intensity: 1 = little to no staining, 2 = staining obvious only at X400, 3 = staining obvious at X100 but not X40, and 4 = staining obvious at X40). An IHC score was calculated by multiplying the values of the staining percentage and intensity.

### Chromatin immunoprecipitation quantitative PCR (ChIP-qPCR) analysis

Sonicated soluble chromatin was incubated with 2 μg of nonspecific control IgG, LSD1, or MBD3 antibodies using the methods described previously in ref. [52]. DNA eluted from reverse crosslink was used for PCR amplification using specific primers for the binding region of these proteins in the *ZRANB1* gene locus. Quantitative real-time PCR was performed using the iQ SYBR Green Supermix and an iCycler iQTM detection system (Bio-Rad) according to the manufacturer's instructions.

### Cleavage under targets and tagmentation (CUT&Tag)

CUT&Tag assay was performed as described previously[38,39]. Briefly, 100,000 cells were washed twice with wash buffer (20 mM HEPES pH 7.5; 150 mM NaCl; 0.5 mM Spermidine; and 1× Protease inhibitor cocktail). About 10 μL Concanavalin A coated magnetic beads (Bangs Laboratories) were added per sample and incubated at RT for 10 min. The unbound supernatant was removed, and the bead-bound cells were resuspended with Dig-wash buffer (20 mM HEPES pH 7.5; 150 mM NaCl; 0.5 mM Spermidine; 1× Protease inhibitor cocktail; 0.05% Digitonin; and 2 mM EDTA) and incubated with a 1:50 dilution of primary antibody or IgG control antibody on a rotating platform overnight at 4 °C. Secondary antibody (Rabbit Anti-Mouse IgG H&L: Abcam, ab611709; Anti-Rabbit IgG antibody, Goat monoclonal: Millipore AP132) was diluted 1:100 in Dig-wash buffer and cells were incubated at RT for 60 min. A 1:100 dilution of pA-Tn5 adapter complex was prepared in Dig-med buffer (0.01% Digitonin; 20 mM HEPES pH 7.5; 300 mM NaCl; 0.5 mM Spermidine; and 1× Protease inhibitor cocktail) and incubated with cells at RT for 1 h. Then cells were resuspended in tagmentation buffer (10 mM MgCl2 in Dig-med Buffer) and incubated at 37 °C for 1 h. DNA was purified using phenol-chloroform-isoamyl alcohol extraction and ethanol precipitation. To amplify libraries, 21 μL DNA was mixed with 2 μL of a universal i5 and barcoded i7 primer. A volume of 25 μL NEBNext HiFi 2× PCR Master mix was added and mixed. The sample was placed in a Thermocycler with a heated lid using the following cycling conditions: 72 °C for 5 min (gap filling); 98 °C for 30 s; 14 cycles of 98 °C for 10 s and 63 °C for 30 s; final extension at 72 °C for 1 min and hold at 8 °C.

### HR and NHEJ reporter assay

Cells were transfected with siControl, siCtiP or si53BP1, shTRABID, Tsin-TRABID-WT, or Tsin-TRABID-C443S separately, and with combinations of HR (pDR-GFP)− or NHEJ (pPEM1-Ad2-EGFP)−reporter constructs and an expression vector for the restriction enzyme I-Sce I. All plasmids used in the GFP-reporter assay were a gift of Zhenkun Lou (Mayo Clinic). U2OS cells integrated with a DR-GFP cassette were used to analyze chromosomal HR efficiency. The GFP expression induced by the positive control plasmid was used to normalize the electroporation efficiency. Cells were grown for 48 h and processed for further flow cytometry analysis.

### Cell synchronization

PC-3 and U2OS cells were treated with nocodazole (100 ng/ml) for 12 h and were released into a regular medium. At the indicated time points after releasing, cells were harvested for cell cycle profiling and western blot analysis using the methods described previously in ref. [20].

### Karyotype analysis

PC-3 cells were treated with 1 μM CPT for 24 h and colcemid for 1 h before harvest. Cells were washed two times in PBS and then resuspended in 0.075 M KCl at 37 °C for 15 min. Cells were fixed with fixative (3:1 methanol:glacial acetic acid) twice, for 15 min each time. Small drops of cell suspension were placed onto a slide surface and stained with Diff-Quick staining for 1 min. Approximately 100 cells with well-spread chromosomes were photographed and analyzed and counted in each group.

### MTS dose-dependent assay and clonogenic assay

For the MTS dose-dependent survival assay, the cells were plated at a density of 1000 cells/well in 96-well plates. After 24 h, the cells were treated with different concentrations of drugs and harvested at 48 h post-treatment. The OD value was read at a wavelength of 490 nm. For the clonogenic survival assay, an appropriate number of cells for different dosages of drugs were plated onto six-well plates. After 24 h, cells were treated with DMSO or different doses of drugs. Twelve days later, colonies were fixed and stained with crystal violet 0.5% (w/v) for 1 h. The number of colonies in each group was counted and analyzed.

### Generation and treatment of prostate cancer xenografts in mice

6-week-old SCID male mice were bred in-house and used for animal experiments. The animal study was approved by the IACUC at Mayo Clinic. All mice were housed under standard conditions at room temperature with a 12 h light/dark cycle with access to food and water ad libitum and maintained under pathogen-free conditions. Carbon dioxide ($CO_2$) inhalation was used for euthanization. For studies with tumors treated with PARP inhibitor olaparib, PC-3 cells ($5 \times 10^6$) infected with lentivirus expressing pTsin-vector, pTsin-TRABID-WT, or pTsin-TRABID-C443S mutant were injected s.c. into the right flank of mice. After xenografts reached a size of ~100 mm³, mice were treated intraperitoneally daily with vehicle or olaparib (50 mg/kg). The volume of xenografts was measured every 5 days for 30 days and calculated using the formula $0.5 \times Length (L) \times Width (W)^2$. The allowed maximal tumor size is 2 cm in any direction based on the institutional tumor production policies. Upon the completion of the measurement, graft tumors were harvested for photography.

### Generation of graphs and statistical analysis

Graphs were generated using GraphPad Prism 8 project (GraphPad, Inc.) or Microsoft Office Excel 2010. All numerical data are presented as mean ± SD as required. Differences between groups were compared by *t*-tests or two-way ANOVA by GraphPad Prism 8 project for Statistical Computing. The following symbols were used to denote statistical significance: n.s. not significant.

### Reporting summary

Further information on research design is available in the Nature Portfolio Reporting Summary linked to this article.

## Data availability

All data is available in the article, Supplementary Information, and source data. The raw data for the western blot generated in this study have been deposited in the Mendeley database (https://data.mendeley.com/datasets/n9txt6y5cj/1). The raw sequencing data for CUT&Tag have been deposited to the National Center for Biotechnology Information (NCBI) Gene Expression Omnibus (GEO) database with the accession code GSE222267. Source data are provided with this paper.

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

## Acknowledgements
This work was supported, in part, by grants from the National Natural Science Foundation of China (82173037 to J.M. and 82102794 to Y.Z.) and the Mayo Clinic Foundation (to H.H.)

## Author contributions
H.H. and L.L. conceived the study. J.M., Y.Z., Z.W., P.P., H.Y., L.L.L., B.W., Y.Y., Y.P., T.L., X.F., S.X., K.W., X.W., Y.J., B.M., Y.F., Y.G., and Q.Y. performed experiments, data collection, and analysis. H.H., L.L., and J.M. supervised the study and wrote the manuscript.

## Competing interests
The authors declare no competing interests.
