## [Peer Review File · Nature Communications]

TRABID overexpression enables synthetic lethality to PARP inhibitor via prolonging 53BP1 retention at double-strand breaksREVIEWER COMMENTS

Reviewer #1 (Remarks to the Author):

In the manuscript entitled "TRABID deubiquitinase overexpression enables synthetic lethality to PARP inhibitor via prolonged retention of 53BP1 at DSB sites", the authors demonstrated that TRABID binds to 53BP1 at endogenous level and regulates 53BP1 retention at DSB sites by deubiquitinating K29-linked polyubiquitination of 53BP1. They further revealed that TRABID overexpression inhibits HR activity and promotes chromosomal instability, which enables synthetic lethality to PARP inhibitor in prostate cancer cells. This group previously reported that SPOP binds to 53BP1 in S phase of the cell cycle and promotes K29-linked polyubiquitination of 53BP1, which leads to 53BP1 extraction from DSB sites. The present study is a good extension for the role of 53BP1 in maintaining the balance between HR and NHEJ. The authors' findings provided new evidence for PTM leading to HR deficiencies and the potential therapeutic use of PARP1 inhibitors in prostate cancer.

Overall, the manuscript is well organized, and provides new insights in the field.

Major concerns:

1. The authors demonstrated TRABID binds the focus-forming region of 53BP1, while SPOP also recognizes the FFR of 53BP1 according to their previous study. Considering TRABID is overexpressed in prostate cancer, it is necessary to figure out if TRABID-53BP1 interaction impairs SPOP binding to 53BP1. According to the current results, the catalytical activity is necessary for TRABID prolonging retention of 53BP1 at DSB sites, but no evidence eliminates the possibility that overexpressed TRABID impairs SPOP binding to 53BP1, which consequently prevents 53BP1 dissociation from DSB sites.

2. In the previous study, the authors demonstrated that SPOP interacts with and mediates 53BP1 polyubiquitination primarily during S phase. In this study, the authors also showed TRABID regulates 53BP1 de-ubiquitination in a cell cycle-dependent manner (Figure 4 j,k,i) but did not conclude in which phase TRABID deubiquitinates 53BP1. This will help understanding how TRABID antagonizes SPOP-mediated 53BP1 polyubiquitination.

Minor points:

1. P 28 Line 623. Figure 3a presents both PC-3 and U2OS cells
2. P 34 Line 684. PC-3 should refer to a, b while U2OS refer to c, d.

Reviewer #2 (Remarks to the Author):

In this paper, the authors describe an additional layer of regulation of 53BP1 that affect the regulation of DNA Double strand break pathway choice. These results showed are solid, and they can convincingly support a model in which the DUB TRABID counteract the Ubiquitination of 53BP1 by SPOP upon DNA damage, affecting the retention of this critical factor. As expected, affecting this retention is accompanied by a plethora of phenotypes in DNA repair assays. Interestingly, the interaction between 53BP1 and TRABID seems constitutive, an issue that it might be worth it include in the discussion. What the authors have uncovered is that TRABID protein levels themselves are controlled at the level of transcription by the NuRD complex in response to DNA damage but also are cell cycle regulated. Furthermore, TRABID levels are affected in several cancer, and here the authors focus in prostate cancer. Strikingly, by impairing recombination TRABID overexpression render cells sensitive to PARPi, suggesting that TRABID levels might be used as a biomarker to determine which patients can benefit of this treatment. Overall, the data are solid, the story is interestingly and the data is able to support the model. The critical part is if the advance in knowledge is enough to grant publication in Nat Comm or not. It is well established in the field that anything that affects 53BP1 recruitment/retention will show the obtained results, including the sensitivity to PARPi. On the plus side, the author show this new

regulatory layer that rely on TRABID, but also how TRABID is itself regulated by the DNA damage response. I find difficult to judge myself if the advance is enough or not, so although I do not enthusiastically promote its publication I am not opposed either if the editor and the others reviewers agree.

There are several interesting questions that are easy to address and that might be worth to explore in a revised version:

1. Is the cell cycle regulation of TRABID also dependent on the NuRD complex or not?
2. DUBs are notoriously promiscuous enzymes, so it is unlikely that all the effect of this protein in the DNA damage response is through 53BP1. What is the effect, at least on DNA damage sensitivity, of TRABID depletion or overexpression in 53BP1 KO cells?

Minor issues:

1. How do the authors explain that TRABID interaction is constitutive? This is strange, as enzymes usually are difficult to IP with their substrates due to the transient nature of their interaction. Do the authors think they form a stable complex?
2. If so, why TRABID depletion has little effect on 53BP1 Ubiquitination in unchallenged cells?
3. Panel 4m is completely unnecessary. I will recommend to eliminate it.

.

Reviewer #3 (Remarks to the Author):

In the manuscript, Ma and colleagues present TRABID retains 53BP1 foci at DSB sites via inhibition of SPOP-dependent ubiquitylation and overexpression of TRABID sensitizes prostate cancer cells to PARP inhibitor. First, they identify TRABID as a candidate of deubiquitinase disturbs 53BP1 foci formation after IR using a small-scale RNAi-based screen. They show TRABID deubiquitylates SPOP-mediated K27-linked ubiquitylated 53BP1 after IR, leading to retention of 53BP1 foci in DSB sites, inhibition of BRCA1/RAD51 foci formation and attenuation of homologous recombination activity. TRABID mRNA levels are regulated by NuRD complex after IR and the levels in unchallenged cells are decreased during S-phase. Finally, they also show TRABID overexpression increases the effect of olaparib treatment to cancer cells and prostate cancer-derived xenograft tumor.

These findings will be of interest to scientists in the field of DSB response and cancer because molecular mechanisms of 53BP1 localization after induction of DSBs is still a hot topic. However, the authors should address several points to strengthen their claims.

Comments:

- 1) Fig. 3b, 3c, 4l, experiments to see ubiquitylation of 53BP1 are performed using a RIPA-like buffer (50 mM Tris-HCl, pH 7.5, 150 mM NaCl, 1% NP40, 0.5% sodium deoxycholate). In this experimental condition, the authors cannot argue the smear bands show 53BP1-(Ub)_n due to the possibility that the immunoprecipitate contains proteins interact with 53BP1. The author should perform the experiments using a denaturing buffer which is capable to dissociate non-covalent protein-protein interaction.
- 2) Fig. 4, the authors find NuRD complex represses the transcription of ZRANB1 after IR. Can the authors explain the molecular mechanisms by which NuRD complex specifically binds to the promotor proximal region of the ZRANB1 gene in response to induction of DSBs? The authors may perform ChIP-seq experiments to see the distribution of NuRD complex after IR in genome-wide.
- 3) Fig. 4l, the authors should investigate whether decreased levels of TRABID in S-phase is dependent on the binding of NuRD complex to the promotor proximal region of the ZRANB1 gene.
- 4) Fig. 6b and 6d, the authors should perform statistical analysis among cells infected with lentivirus expressing EV, TRABID-WT and TRABID-C443S in the presence of olaparib after normalization to colony number from cells untreated with olaparib.
- 5) Fig. 5f, overexpression of TRABID inhibits the recruitment of BRCA1 and RAD51 at DSBs sites. To further validate TRABID-dependent deubiquitination of 53BP1 compromises localization of HR-

related proteins to DSB sites, the authors should test whether TRABID overexpression does not affect foci formation of HR-related proteins following IR in 53BP1-deficient cells.

RESPONSE TO REVIEWERS' COMMENTS

We very much thank the Reviewers for the positivity and insightful comments for the improvement of our manuscript.

Reviewer #1 (Remarks to the Author):

In the manuscript entitled “TRABID deubiquitinase overexpression enables synthetic lethality to PARP inhibitor via prolonged retention of 53BP1 at DSB sites”, the authors demonstrated that TRABID binds to 53BP1 at endogenous level and regulates 53BP1 retention at DSB sites by deubiquitinating K29-linked polyubiquitination of 53BP1. They further revealed that TRABID overexpression inhibits HR activity and promotes chromosomal instability, which enables synthetic lethality to PARP inhibitor in prostate cancer cells. This group previously reported that SPOP binds to 53BP1 in S phase of the cell cycle and promotes K29-linked polyubiquitination of 53BP1, which leads to 53BP1 extraction from DSB sites. The present study is a good extension for the role of 53BP1 in maintaining the balance between HR and NHEJ. The authors’ findings provided new evidence for PTM leading to HR deficiencies and the potential therapeutic use of PARP1 inhibitors in prostate cancer.

Overall, the manuscript is well organized, and provides new insights in the field.

Response: We thank the Reviewer for recognizing the novelty and significance of our finding. We also thank the Reviewer for the insightful suggestions that have helped us improve our manuscript significantly.

Major concerns:

1. The authors demonstrated TRABID binds the focus-forming region of 53BP1, while SPOP also recognizes the FFR of 53BP1 according to their previous study. Considering TRABID is overexpressed in prostate cancer, it is necessary to figure out if TRABID-53BP1 interaction impairs SPOP binding to 53BP1. According to the current results, the catalytical activity is necessary for TRABID prolonging retention of 53BP1 at DSB sites, but no evidence eliminates the possibility that overexpressed TRABID impairs SPOP binding to 53BP1, which consequently prevents 53BP1 dissociation from DSB sites.

Response: We thank the Reviewer for raising these great points. We agree that it is important to determine whether TRABID-53BP1 interaction impairs SPOP binding to 53BP1. To experimentally address this concern, we performed co-IP assay. We found that neither expression of WT TRABID nor the catalytically inactive mutant C433S had any obvious effect on SPOP-53BP1 interaction (**Supplementary Fig. 2h**). These data indicate that TRABID antagonizes SPOP-mediated 53BP1 polyubiquitination without affecting SPOP binding to 53BP1.

2. In the previous study, the authors demonstrated that SPOP interacts with and mediates 53BP1 polyubiquitination primarily during S phase. In this study, the authors also showed TRABID regulates 53BP1 de-ubiquitination in a cell cycle-dependent manner (Figure 4 j,k,i) but did not

conclude in which phase TRABID deubiquitinates 53BP1. This will help understanding how TRABID antagonizes SPOP-mediated 53BP1 polyubiquitination.

Response: This is an excellent point. To determine the cell cycle phase in which TRABID-mediated deubiquitination of 53BP1 occurs, we synchronized PC-3 and U2OS cells by nocodazole treatment and release (**Fig. 4k**). We found that TRABID protein expression varied during the cell cycle with the lowest expression occurring in the S phase (**Fig. 4l**). Similar to protein expression, we further demonstrated that both *ZRANB1* mRNA expression also oscillated during the cell cycle with low expression at the S phase in PC-3 cells (**Fig. 4m-p**). In agreement with the expression pattern of TRABID during the cell cycle, we found that K29-linked polyubiquitinated of 53BP1 also peaked in the S phase of PC-3 cells; however, depletion of TRABID delayed the process of 53BP1 deubiquitination during the cell cycle (**Fig. 4q**). These data indicate that TRABID deubiquitinates 53BP1 during the S phase.

Minor points:

1. P 28 Line 623. Figure 3a presents both PC-3 and U2OS cells

Response: We thank the Reviewer for mentioning this point. The error has been corrected in the revised manuscript.

2. P 34 Line 684. PC-3 should refer to a, b while U2OS refer to c, d.

Response: We thank the Reviewer for pointing this out. The error has been corrected in the revised manuscript.

Reviewer #2 (Remarks to the Author):

In this paper, the authors describe an additional layer of regulation of 53BP1 that affect the regulation of DNA Double strand break pathway choice. The results showed are solid, and they can convincingly support a model in which the DUB TRABID counteract the Ubiquitination of 53BP1 by SPOP upon DNA damage, affecting the retention of this critical factor. As expected, affecting this retention is accompanied by a plethora of phenotypes in DNA repair assays. Interestingly, the interaction between 53BP1 and TRABID seems constitutive, an issue that it might be worth it include in the discussion. What the authors have uncovered is that TRABID protein levels themselves are controlled at the level of transcription by the NuRD complex in response to DNA damage but also are cell cycle regulated. Furthermore, TRABID levels are affected in several cancer, and here the authors focus in prostate cancer. Strikingly, by impairing recombination TRABID overexpression render cells sensitive to PARPi, suggesting that TRABID levels might be used as a biomarker to determine which patients can benefit of this treatment.

Overall, the data are solid, the story is interestingly and the data is able to support the model. The critical part is if the advance in knowledge is enough to grant publication in Nat Comm or not. It is well established in the field that anything that affects 53BP1 recruitment/retention will show the obtained results, including the sensitivity to PARPi. On the plus side, the authors show this new regulatory layer that rely on TRABID, but also how TRABID is itself regulated by the DNA

damage response. I find difficult to judge myself if the advance is enough or not, so although I do not enthusiastically promote its publication I am not opposed either if the editor and the others reviewers agree.

Response: We thank the Reviewer for the positive comments on our manuscript and great suggestions.

There are several interesting questions that are easy to address and that might be worth to explore in a revised version:

1. Is the cell cycle regulation of TRABID also dependent on the NuRD complex or not?

Response: We thank the Reviewer for raising this excellent point. In synchronized PC-3 and U2OS cells (**Fig. 4k**), we demonstrated that TRABID protein expression varied during the cell cycle with the lowest expression occurring in the S phase (**Fig. 4l**). Similar to protein expression, we further demonstrated that both *ZRANB1* mRNA expression also oscillated during the cell cycle with low expression at the S phase in PC-3 cells (**Fig. 4m-p**). Previously studies showed that the level of the NuRD complex components LSD1 and MBD3 level peaks in early S phase (Wang et al., *Signal Transduct Target Ther*, 2022) (*see all the references at the end of the rebuttal letter*). We demonstrated that knockdown of LSD1 or MBD3 abolished the oscillation of TRABID expression (**Fig. 4m-p**). These data suggest that the components of the NuRD complex, such as LSD1 and MBD3, play an important role in the cell cycle-dependent regulation of TRABID expression.

2. DUBs are notoriously promiscuous enzymes, so it is unlikely that all the effect of this protein in the DNA damage response is through 53BP1. What is the effect, at least on DNA damage sensitivity, of TRABID depletion or overexpression in 53BP1 KO cells?

Response: We thank the Reviewer for this great question. To experimentally address this concern, we stably expressed TRABID WT and the catalytically inactive mutant C443S (a negative control) into PC-3 and U2OS cell lines (**Supplementary Fig. 4a**). As expected, TRABID WT overexpression prolonged 53BP1 retention at DSB sites; however, expression of TRABID WT but not C443S significantly decreased the number of IRIF of HR mediators BRCA1 and RAD51 (**Fig. 5d-g and Supplementary Fig. 4b-i**). Of note, knockdown of 53BP1 abolished TRABID-mediated decrease in BRCA1 and RAD51 foci (**Fig. 5d-g and Supplementary Fig. 4b-i**), suggesting that TRABID overexpression-mediated inhibition of HR protein foci formation is 53BP1-dependent. Moreover, a dose course study in both PC-3 and U2Os cells revealed that the IC50 of the PARP inhibitor olaparib in TRABID WT-expressing cells was much lower compared with EV control and TRABID C443S-expressing cells; however, the effect of TRABID WT was abolished by 53BP1 knockdown (**Fig. 6a, b and Supplementary Fig. 4a**). These data suggest TRABID overexpression influences DNA damage response mainly through 53BP1.

Minor issues:

1. How do the authors explain that TRABID interaction is constitutive? This is strange, as enzymes usually are difficult to IP with their substrates due to the transient nature of their interaction. Do the authors think they form a stable complex?

Response: This is an excellent point. There are several findings supporting the DUB enzymes can bind to substrates constitutively such as USP10-p53 (Yuan et al., Cell, 2010), DUB3-BRD4 (Jin et al., Mol Cell, 2018), USP4-TβRI (Zhang et al., Nat Cell Biol, 2012), OTUD1-SMAD7 (Zhang et al., Nat Commun, 2017), UCHL1-EGFR (Bi et al., Sci Adv, 2020), CSN5-PD-L1 (Lim et al., Cancer Cell, 2016), ATX3-beclin 1 (Ashkenazi et al., Nature, 2017), as well as the interaction of the DUB enzyme TRABID with other proteins such as TRABID-EZH2 interaction (Zhang et al., Cell Rep, 2018). Our results showed that both ectopically expressed and endogenous TRABID interacted with 53BP1 in 293T cells and PC-3 prostate cancer cells (**Fig. 2a, b**), indicating that TRABID can form a stable complex with 53BP1.

2. If so, why TRABID depletion has little effect on 53BP1 Ubiquitination in unchallenged cells?

Response: We thank the Reviewer for raising this great point. We agree that TRABID depletion has little effect on 53BP1 ubiquitination in cells without IR treatment. However, as reported previously by us and others (Kristariyanto et al., Mol Cell, 2015; Michel et al., Mol Cell, 2015; Wang et al., Sci Adv, 2021), 53BP1 can only be polyubiquitinated after IR treatment and SPOP specifically promotes K29-linked polyubiquitination of 53BP1 under DNA damage conditions. We provide evidence in the present study that TRABID specifically binds and deubiquitinates K29-linked ubiquitin chains of 53BP1, antagonizing SPOP-mediated 53BP1 polyubiquitination. Our results further confirmed that expression of the wild type (WT) TRABID but not the catalytically inactive mutant (C443S) induced a marked decrease in 53BP1 polyubiquitination upon IR treatment (**Supplementary Fig. 2a**). Importantly, expression of TRABID had little or none effect on 53BP1 deubiquitination in cells expressing SPOP F133V (**Supplementary Fig. 2a**). We further showed that TRABID also deubiquitinated K29-linked polyubiquitination of 53BP1 in SPOP WT but not F133V mutant cells (**Fig. 3b and Supplementary Fig. 2b**). These data indicate that TRABID mediates deubiquitination of K29-linked polyubiquitination of 53BP1, a substrate that can only be available in the presence of DNA damage and a functional SPOP in challenged cells.

3. Panel 4m is completely unnecessary. I will recommend to eliminate it.

Response: We agree with the Reviewer. We have now eliminated it in the revised version.

Reviewer #3 (Remarks to the Author):

In the manuscript, Ma and colleagues present TRABID retains 53BP1 foci at DSB sites via inhibition of SPOP-dependent ubiquitylation and overexpression of TRABID sensitizes prostate cancer cells to PARP inhibitor. First, they identify TRABID as a candidate of deubiquitinase disturbs 53BP1 foci formation after IR using a small-scale RNAi-based screen. They show TRABID deubiquitylates SPOP-mediated K27-linked ubiquitylated 53BP1 after IR, leading to retention of 53BP1 foci in DSB sites, inhibition of BRCA1/RAD51 foci formation and attenuation of homologous recombination activity. TRABID mRNA levels are regulated by

NuRD complex after IR and the levels in unchallenged cells are decreased during S-phase. Finally, they also show TRABID overexpression increases the effect of olaparib treatment to cancer cells and prostate cancer-derived xenograft tumor.

These findings will be of interest to scientists in the field of DSB response and cancer because molecular mechanisms of 53BP1 localization after induction of DSBs is still a hot topic. However, the authors should address several points to strengthen their claims.

Response: We very much appreciate the Reviewer for the positivity and enthusiasm about our study.

Comments:

1) Fig. 3b, 3c, 4l, experiments to see ubiquitylation of 53BP1 are performed using a RIPA-like buffer (50 mM Tris-HCl, pH 7.5, 150 mM NaCl, 1% NP40, 0.5% sodium deoxycholate). In this experimental condition, the authors cannot argue the smear bands show 53BP1-(Ub)_n due to the possibility that the immunoprecipitate contains proteins interact with 53BP1. The author should perform the experiments using a denaturing buffer which is capable to dissociate non-covalent protein-protein interaction.

Response: We thank the Reviewer for these helpful suggestions. As the Reviewer suggested, we repeated all these ubiquitination assays using nickel-nitrilotriacetic acid (Ni-NTA) beads in the denaturing lysis buffer containing guanidine-HCl. Our new data (**Fig. 3b, 3c and 4q**) are consistent with our original data, which are now moved to the Supplementary Figures (**Supplementary Fig. 2b and 2c**).

2) Fig. 4, the authors find NuRD complex represses the transcription of ZRANB1 after IR. Can the authors explain the molecular mechanisms by which NuRD complex specifically binds to the promoter proximal region of the ZRANB1 gene in response to induction of DSBs? The authors may perform CHIP-seq experiments to see the distribution of NuRD complex after IR in genome-wide.

Response: We thank the Reviewer for raising these excellent points. We noticed that an enzyme-tethering strategy termed Cleavage Under Targets and Tagmentation (CUT&Tag) is a very useful epigenomic profiling method, which can provide efficient high-resolution sequencing libraries with exceptionally low background for profiling diverse chromatin components (Kaya-Okur et al., Nat Protoc, 2020; Kaya-Okur et al., Nat Commun, 2019) (*see all the references at the end of the rebuttal letter*). We therefore performed CUT&Tag profiling of LSD1 and MBD3, two key components of the NuRD complex in the presence or absence of IR treatment and examined the genome-wide distribution of these NuRD complex proteins after IR. We demonstrated that IR treatment induced a genome-wide increase in the distribution of LSD1 and MBD3 in PC-3 cells, although the increase was in a very limited scope (**Fig. 4c**). Importantly, we found that DNA damage substantially increased the occupancy of these repressor proteins in the promoter proximal region of the ZRANB1 gene (**Fig. 4d**).

It is worth noting that these results are consistent with our finding that DNA damage-induced repression of ZRANB1 mRNA expression was completely abolished by knockdown of each of

these two components of the NuRD complex (**Fig. 4g-j**). These data highlight the importance of DNA damage-induced engagement of LSD1 and MBD3 proteins in ZRANB1 expression repression and the coordinated action of these two components of the NuRD complex.

3) Fig. 4l, the authors should investigate whether decreased levels of TRABID in S-phase is dependent on the binding of NuRD complex to the promotor proximal region of the ZRANB1 gene.

Response: We thank the Reviewer for raising this great point. As mentioned above, our new data showed that both TRABID protein and mRNA expression levels oscillated during the cell cycle, with decreased expression of TRABID at the S phase in PC-3 cells (**Fig. 4m-p**). Moreover, as reported previously (Wang et al., *Signal Transduct Target Ther*, 2022), the levels of both LSD1 and MBD3, two key components of the NuRD complex peaked in early S phase, but most importantly, knockdown of LSD1 or MBD3 abolished downregulation of TRABID expression in the S phase (**Fig. 4m-p**), supporting the notion that downregulation of TRABID expression during the cell cycle is NuRD complex-mediated.

4) Fig. 6b and 6d, the authors should perform statistical analysis among cells infected with lentivirus expressing EV, TRABID-WT and TRABID-C443S in the presence of olaparib after normalization to colony number from cells untreated with olaparib.

Response: We thank the Reviewer for the great suggestion. We have performed statistical analysis and updated the new results in **Fig. 6d** and **Fig. 6f** in the revised manuscript.

5) Fig. 5f, overexpression of TRABID inhibits the recruitment of BRCA1 and RAD51 at DSBs sites. To further validate TRABID-dependent deubiquitination of 53BP1 compromises localization of HR-related proteins to DSB sites, the authors should test whether TRABID overexpression does not affect foci formation of HR-related proteins following IR in 53BP1-deficient cells

Response: We thank the Reviewer for these excellent suggestions. As the Reviewer suggested, we stably expressed TRABID WT and the catalytically inactive mutant C443S (a negative control) into PC-3 and U2OS cell lines (**Supplementary Fig. 4a**). Consistent with the observation that TRABID WT expression prolonged 53BP1 retention at DSB sites (**Fig. 3d-f and Supplementary Fig. 2e-g**), expression of TRABID WT but not C443S significantly decreased the number of IRIF of HR mediators BRCA1 and RAD51 (**Fig. 5d-g and Supplementary Fig. 4b-i**). Of note, knockdown of 53BP1 abolished TRABID-mediated decrease in BRCA1 and RAD51 foci (**Fig. 5d-g and Supplementary Fig. 4b-i**), suggesting that TRABID overexpression-mediated inhibition of HR protein foci formation is 53BP1-dependent.

References

- Ashkenazi, A., Bento, C.F., Ricketts, T., Vicinanza, M., Siddiqi, F., Pavel, M., Squitieri, F., Hardenberg, M.C., Imarisio, S., Menzies, F.M., *et al.* (2017). Polyglutamine tracts regulate beclin 1-dependent autophagy. *Nature* 545, 108-111.
- Bi, H.L., Zhang, X.L., Zhang, Y.L., Xie, X., Xia, Y.L., Du, J., and Li, H.H. (2020). The deubiquitinase UCHL1 regulates cardiac hypertrophy by stabilizing epidermal growth factor receptor. *Sci Adv* 6, eaax4826.

Jin, X., Yan, Y., Wang, D., Ding, D., Ma, T., Ye, Z., Jimenez, R., Wang, L., Wu, H., and Huang, H. (2018). DUB3 Promotes BET Inhibitor Resistance and Cancer Progression by Deubiquitinating BRD4. *Mol Cell* **71**, 592-605 e594.

Kaya-Okur, H.S., Janssens, D.H., Henikoff, J.G., Ahmad, K., and Henikoff, S. (2020). Efficient low-cost chromatin profiling with CUT&Tag. *Nat Protoc* **15**, 3264-3283.

Kaya-Okur, H.S., Wu, S.J., Codomo, C.A., Pledger, E.S., Bryson, T.D., Henikoff, J.G., Ahmad, K., and Henikoff, S. (2019). CUT&Tag for efficient epigenomic profiling of small samples and single cells. *Nat Commun* **10**, 1930.

Kristariyanto, Y.A., Abdul Rehman, S.A., Campbell, D.G., Morrice, N.A., Johnson, C., Toth, R., and Kulathu, Y. (2015). K29-selective ubiquitin binding domain reveals structural basis of specificity and heterotypic nature of k29 polyubiquitin. *Mol Cell* **58**, 83-94.

Lim, S.O., Li, C.W., Xia, W., Cha, J.H., Chan, L.C., Wu, Y., Chang, S.S., Lin, W.C., Hsu, J.M., Hsu, Y.H., *et al.* (2016). Deubiquitination and Stabilization of PD-L1 by CSN5. *Cancer Cell* **30**, 925-939.

Michel, M.A., Elliott, P.R., Swatek, K.N., Simicek, M., Pruneda, J.N., Wagstaff, J.L., Freund, S.M., and Komander, D. (2015). Assembly and specific recognition of k29- and k33-linked polyubiquitin. *Mol Cell* **58**, 95-109.

Wang, D., Ma, J., Botuyan, M.V., Cui, G., Yan, Y., Ding, D., Zhou, Y., Krueger, E.W., Pei, J., Wu, X., *et al.* (2021). ATM-phosphorylated SPOP contributes to 53BP1 exclusion from chromatin during DNA replication. *Sci Adv* **7**.

Wang, Y., Huang, Y., Cheng, E., Liu, X., Zhang, Y., Yang, J., Young, J.T.F., Brown, G.W., Yang, X., and Shang, Y. (2022). LSD1 is required for euchromatic origin firing and replication timing. *Signal Transduct Target Ther* **7**, 102.

Yuan, J., Luo, K., Zhang, L., Cheville, J.C., and Lou, Z. (2010). USP10 regulates p53 localization and stability by deubiquitinating p53. *Cell* **140**, 384-396.

Zhang, L., Zhou, F., Drabsch, Y., Gao, R., Snaar-Jagalska, B.E., Mickanin, C., Huang, H., Sheppard, K.A., Porter, J.A., Lu, C.X., *et al.* (2012). USP4 is regulated by AKT phosphorylation and directly deubiquitylates TGF-beta type I receptor. *Nat Cell Biol* **14**, 717-726.

Zhang, P., Xiao, Z., Wang, S., Zhang, M., Wei, Y., Hang, Q., Kim, J., Yao, F., Rodriguez-Aguayo, C., Ton, B.N., *et al.* (2018). ZRANB1 Is an EZH2 Deubiquitinase and a Potential Therapeutic Target in Breast Cancer. *Cell Rep* **23**, 823-837.

Zhang, Z., Fan, Y., Xie, F., Zhou, H., Jin, K., Shao, L., Shi, W., Fang, P., Yang, B., van Dam, H., *et al.* (2017). Breast cancer metastasis suppressor OTUD1 deubiquitinates SMAD7. *Nat Commun* **8**, 2116.

REVIEWERS' COMMENTS

Reviewer #1 (Remarks to the Author):

All previous comments are well addressed. No further questions.

Reviewer #2 (Remarks to the Author):

The authors have addressed all the points raised in my review. As mentioned before, the manuscript data fully agree with the authors' model. Thus, I am happy to support its publication in the current format.

Reviewer #3 (Remarks to the Author):

The revised manuscript by Ma et al. has addressed all the comments. I would recommend it for publication in Nature Communications.

RESPONSE TO REVIEWERS' COMMENTS

We very much thank the Reviewers for the positivity and insightful comments for our revised manuscript. We are very glad that they are all supportive for the publication of our manuscript in *Nature Communications*.

Reviewer #1 (Remarks to the Author):

All previous comments are well addressed. No further questions.

Response: We thank the Reviewer for the enthusiasm and recommendation for the publication of our manuscript without any additional changes.

Reviewer #2 (Remarks to the Author):

The authors have addressed all the points raised in my review. As mentioned before, the manuscript data fully agree with the authors' model. Thus, I am happy to support its publication in the current format.

Response: We thank the Reviewer for the enthusiasm and recommendation for the publication of our manuscript without any additional changes.

Reviewer #3 (Remarks to the Author):

The revised manuscript by Ma et al. has addressed all the comments. I would recommend it for publication in *Nature Communications*.

Response: We thank the Reviewer for the enthusiasm and recommendation for the publication of our manuscript without any additional changes.